# Enhanced Carbofuran Degradation Using Immobilized and Free Cells of *Enterobacter* sp. Isolated from Soil

**DOI:** 10.3390/molecules25122771

**Published:** 2020-06-16

**Authors:** Mohammed Umar Mustapha, Normala Halimoon, Wan Lutfi Wan Johari, Mohd. Yunus Abd Shukor

**Affiliations:** 1Desert Research Monitoring and Control Centre, Yobe State University, Damaturu P.M.B 1144, Nigeria; umardrc@gmail.com; 2Department of Environment, Faculty of Forestry and Environment, Universiti Putra Malaysia, UPM Serdang, Selangor 43400, Malaysia; wanlutfi@upm.edu.my; 3Department of Biochemistry, Faculty of Biotechnology and Biomolecular Sciences, Universiti Putra Malaysia, UPM Serdang, Selangor 43400, Malaysia; mohdyunus@upm.edu.my

**Keywords:** biodegradation, carbofuran, immobilization, reusability

## Abstract

Extensive use of carbofuran insecticide harms the environment and human health. Carbofuran is an endocrine disruptor and has the highest acute toxicity to humans than all groups of carbamate pesticides used. Carbofuran is highly mobile in soil and soluble in water with a lengthy half-life (50 days). Therefore, it has the potential to contaminate groundwater and nearby water bodies after rainfall events. A bacterial strain BRC05 was isolated from agricultural soil characterized and presumptively identified as *Enterobacter* sp. The strain was immobilized using gellan gum as an entrapment material. The effect of different heavy metals and the ability of the immobilized cells to degrade carbofuran were compared with their free cell counterparts. The results showed a significant increase in the degradation of carbofuran by immobilized cells compared with freely suspended cells. Carbofuran was completely degraded within 9 h by immobilized cells at 50 mg/L, while it took 12 h for free cells to degrade carbofuran at the same concentration. Besides, the immobilized cells completely degraded carbofuran within 38 h at 100 mg/L. On the other hand, free cells degraded the compound in 68 h. The viability of the freely suspended cell and degradation efficiency was inhibited at a concentration greater than 100 mg/L. Whereas, the immobilized cells almost completely degraded carbofuran at 100 mg/L. At 250 mg/L concentration, the rate of degradation decreased significantly in free cells. The immobilized cells could also be reused for about nine cycles without losing their degradation activity. Hence, the gellan gum-immobilized cells of *Enterobacter* sp. could be potentially used in the bioremediation of carbofuran in contaminated soil.

## 1. Introduction

Pesticide use in agriculture can cause undesirable effects on humans and the natural environment, and different types of toxic organic compounds are being released accidentally or intentionally into the environment [1]. Pesticides are introduced frequently in large volumes via different means [2]. Carbofuran is among the most frequently used insecticides and has been broadly used for the control of insects that destroy crops, such as, tomatoes, cabbages, rice, strawberries, and many other crops. Almost all pesticides belonging to carbamates are synthetic and are widely used all over the globe due to their broad specificity against several pest’s organisms [3]. Though pesticides have been developed with the concept of the target organism, however, non-target organisms are often affected badly by their application [4]. Carbofuran is a strong inhibitor of acetyl-cholinesterase and has great toxicity, causing health-related disorders [5]. Though carbofuran is chemically unstable as a result of easy hydrolysis in the environment, carbofuran residues are mostly discovered in groundwater because of its extensive use and high mobility in soils [6]. Therefore, their removal from the environment is essential [7,8]. To reduce the environmental and public health risks associated with carbofuran use, it is crucial to develop rapid and effective approaches to remove the compound or reduce their concentrations from the environment [1,9].

Conventional approaches for mitigating the impact of pesticide contamination are less-effective and costly [10]. Among the several techniques available are the biological method, which is based on the catabolic activity of pesticide-degrading bacteria, which seems to be the most promising and effective strategy. Recently, immobilization methods have received a great deal of attention from stakeholders [11]. Cell immobilization is an effective biological technique for the removal of pollutants due to improved operational immobility, higher biomass loading, and enriched biodegradation rates [12]. Degradation of pesticides, such as carbofuran, using free cells has been a challenge due to some limitations in their application for bioremediation, which includes cell separation, substrate inhibition, and difficulty in isolating a strain that can withstand high concentration of carbofuran and possible degradation within a shortest possible time. Therefore, to overcome these challenges, immobilized cell technology can be applied for the long-term stabilization of different pollutant-degrading microbes [13,14,15]. One of the most important aspects of bioremediation is maintaining high biomass of bacterial populations and increase the survival and retention of the bioremediation agents in the polluted areas. Immobilized cells have the possibility of retaining a high cell density, the prevention of cell washout, even at high dilution rates, as well as repeated use of cells, and better protection of cells from harsh environments, which make it have significant benefits over free cells. It has been reported that immobilized cells are much more tolerant of perturbations in the reaction environment and less susceptible to toxic substances, making immobilized cell systems particularly attractive for the degradation of toxic compounds like pesticides [16].

Cell immobilization refers to physical detention or localization of viable microbial cells to a certain distinct area of space so as to limit their free movement with simultaneous preservation of their viability and catalytic functions. It involves two processes—physical retention (inclusion membrane and entrapment) and chemical bonding, for instance, biofilm formation [17]. The technique may use the natural capability of microbes to form biofilms on the surface of various materials, which is commonly observed in the environment. There are five main techniques of immobilization, which are adsorption, binding on a surface (electrostatic or covalent), flocculation (natural or artificial), entrapment, and encapsulation [18]. Several materials are used for immobilization of cells, including inorganic, such as clays, silicates, glass, and ceramics, as well as organic, such as cellulose, starch, dextran, agarose, alginate, chitin, collagen, keratin, etc. Immobilization significantly reduces the costs of bioremediation processes and also increases their efficiency. The immobilized cells could be reused over and over, hence decreasing the costly processes of recycling and recovery of cells [19,20]. Moreover, the immobilized microbial cells may offer many benefits, such as high mechanical strength, high resistance to toxic chemicals, and high metabolic activity, due to high biomass concentrations and dispersal blockades within the biofilm, as well as stopping wash out of cells from the beads [21]. The study aimed at isolating bacteria capable of degrading carbofuran and immobilizing the bacterium on a natural carrier (gellan gum) to produce a metabolically active, biological agent for the biodegradation of carbofuran in polluted soil.

## 2. Results and Discussions

### 2.1. Isolation and Identification

The carbofuran-degrading bacterium was isolated using enrichment methods. The strain BRC05 was isolated in soil and tested for its ability to degrade carbofuran by using the compound as a source of nitrogen or carbon. The bacterium was motile, rod-shaped, and a Gram-negative strain with the flagellum. Based on the biochemistry performed, the isolate was citrate-positive, catalase-negative, oxidase, and non-spore forming bacteria. 16S rRNA gene sequence analysis was done to compare with related sequences available in the NCBI GenBank, and the strain BRC05 showed high similarity with the genus *Enterobacter* sp. The sequence of the amplified DNA showed a high similarity of 97% with the genius *Enterobacter* sp., and the top ten Blast search of closely related species together with their accession numbers are presented in Table 1.

Max score is highest alignment score calculated from the sum of the rewards for matched nucleotides, Total score is the sum of alignment scores of all segments from the same subject sequence E-Value: the number of alignments expected by chance with the calculated score or better. The expect value is the default sorting metric; for significant alignments the E value should be very close to zero. Ident: is the highest percent identity for a set of aligned segments to the same subject sequence.

### 2.2. Degradation Studies for Enterobacter sp. Strain BRC05 by Comparison between Immobilized Cell and Freely-Suspended Cell

The experiment showed the degradation of carbofuran at different time intervals and concentration by free cells and immobilized cells of *Enterobacter* sp. strain BRC05. The immobilized and free bacteria at optimized conditions were tested for their carbofuran-degrading activity, and the microbial growth was inhibited at higher concentrations of carbofuran. At different initial carbofuran concentrations, both free and immobilized bacteria demonstrated similar carbofuran-degrading activity, and carbofuran was degraded completely in the first few hours of incubation. As the concentration of carbofuran increased, the immobilized cells exhibited a faster degrading ability than the free cells.

Figure 1a–e displays the results for the degradation of carbofuran by free and immobilized cells at different times and concentrations. As a negative control, empty beads without bacteria were used at the same carbofuran concentrations. There was no degradation observed in the negative control beads at the end of the experiment. At 50 mg/L carbofuran concentration (Figure 1a), the immobilized cells were able to degrade carbofuran completely within 9 h when compared to the free cell, which took 12 h to degrade carbofuran. With the increased concentration of carbofuran up to 250 mg/L, the rate of degradation decreased in free cells. The immobilized cells completely degraded carbofuran within 15 and 38 h at 50–100 mg/L. On the other hand, free cells degraded the compound in 14 to 68 h at concentrations 50 and 100 mg/L, respectively (Figure 1b,c). The biodegradation activity of carbofuran by immobilized and free cells at concentrations of 150 and 200 mg/L was somewhat inhibited (Figure 1d,e).

The isolate was incapable of degrading carbofuran completely at these levels, while, at 250 mg/L, the degradation of carbofuran was very minimum for free cells after 160 and 190 h, respectively. Higher concentrations of gellan gum or cell loadings or larger bead sizes could create diffusional limitations, such as decreased oxygen availability to the immobilized cells, which might lead to a reduction in some aspects of cellular activity and reduced degradation capacity. This limitation could be overcome by using smaller beads size or more porous polymer like gellan gum, as well as an appropriate number of beads. Previous studies have shown that immobilization of cells can significantly increase the removal efficiency of different pesticides compared with free cells. Immobilized *Burkholderia cepacia* PCL3 degrades carbofuran at higher concentrations [9]. In another study by Plangklang et al. [16], the strain PCL3 in the free-cell form was able to degrade carbofuran phenol at low concentrations and was inhibited at a high concentration, while the immobilized cells of PCL3 were not inhibited even at higher concentrations of carbofuran phenol.

Similar results were reported by Fareed et al. [15], where immobilized cells of *Enterobacter cloacae* strain TA7 effectively degraded in *N*-methyl carbamates pesticides more rapidly than freely suspended cells counterpart. At 300 mg/L of phenol, the freely suspended cells showed a lower specific degradation rate, whereas, the immobilized cells demonstrated a higher phenol degradation rate [22]. *Pseudomonas putida* strain (CCRC14365) was able to degrade 600 mg/L of phenolic compounds in the free cells system and up to 1000 mg/L with immobilized cells [23]. This further justified that immobilized cells were efficient over time in carbofuran-degrading activities than freely-suspended cells, as shown in the present study. In another study, enhanced degradation of carbofuran phenol was achieved by the immobilized cells of *Klebsiella pneumoniae* as compared with its free cells counterpart [21]. Gellan gum immobilized *Acinetobacter* sp. completely degraded phenol within 108, 216, and 240 h at 1100, 1500, and 1900 mg/L concentration of phenol, while the freely suspended cells took 240 h to completely degrade phenol at 1100 mg/L as the phenol-degrading activity of the free bacteria was inhibited at higher concentrations of the phenolic compound [24].

In another study, phenol degradation by free and immobilized cells of *Bacillus cereus* was similar at lower concentrations of 100 to 1000 mg/L. While almost 50% of 2000 mg/L was degraded within 26 and 36 days. This showed the enhanced degradation efficiency of the immobilized cells, which was higher at higher concentrations than free cells [25]. Besides, Thatheyus et al. [26] reported enhanced degradation of acrylamide by immobilized cells of *Pseudomonas aeruginosa*, where degradation started within 24 h of incubation, whereas it took the free cells 48 h to show degradation. The higher degradation rate of crude oil was also achieved by using immobilized bacterial consortium [27,28]. Additionally, an immobilized cell of *Pseudomonas putida* was reported to have degraded higher concentrations of chlorpyrifos as compared to its free cells [29].

### 2.3. Effect of Heavy Metals on the Degradation of Carbofuran by Enterobacter sp. Strain BRC05

The effect of heavy metals on the degradation of carbofuran, by immobilized and free cells, was checked by culturing in the MSM medium at 37 °C in the presence of different heavy metals. One part per million (1 mg/L) of heavy metals, such as copper (Cu), chromium (Cr), mercury (Hg), cadmium (Cd), arsenic (As), zinc (Zn), lead (Pb), and nickel (Ni), were tested, and the effects of the metals between free and immobilized cells were compared, as shown in Figure 2. A control experiment was prepared without the addition of heavy metals into the medium. The degradation was strongly inhibited by mercury and copper, at 1 mg/L, while the inhibitory effect of lead and chromium at the same concentration was negligible. In contrast, cadmium did not affect the degradation at the concentration used. The degradation of carbofuran did not take place in the control flask by free cells (*p* ≥ 0.05). In immobilized cells, Hg and Cu inhibited carbofuran-degrading activities; however, the degradation was higher compared with the free cells (*p* ≤ 0.05). Hence, different concentrations of Hg and Cu were used for the degradation of carbofuran. The degradation was higher in immobilized than the free cells at the same concentration. The degradation of carbofuran was far lesser in free cells compared to the performance of the immobilized cells. Some heavy metals are essential to living cells, while others may be harmful to living cells even at reduced concentrations. Some heavy metals are readily metabolized and assist in metabolic activities, such as detoxification, assimilation, and methylation. Heavy metals affect the degradation of the pesticides in soil by influencing the microbial actions in the environment [30]. The presence of heavy metals in polluted areas is one of the main limiting factors for bioremediation because several organisms might not tolerate high concentrations of metals; therefore, their ability to degrade pollutants might reduce due to the effect of metals [31], and consequently, the need to assess the effect of heavy metals on the biodegradation of organic compounds. Many researchers reported the effects of heavy metal on the microbial growth; yet, there is very limited information concerning their effects on the degradation of pesticides [22,32].

The result obtained agreed with the work of Hong et al. [33] in which the addition of lead (Pb) had an insignificant inhibitory effect on the degradation of dibenzofuran, and mercury (Hg) highly inhibited the degradation. The free cells were inhibited easily by the heavy metals, hence reducing the rate of degradation through distressing the cells’ membrane structure, whereas the immobilized cells were protected by the beads gel and less affected by the heavy metal, therefore, decreasing the chance of being inhibited by the heavy metals [3,33]. The existence of heavy metals at the polluted site is frequently the main hindering factor for effective bioremediation as the microbial population could not withstand a high concentration of heavy metals, and thus, their ability to degrade the target compounds may decrease [11].

### 2.4. Effect of Mercury (Hg) on the Degradation of Carbofuran by Immobilized Cells of Enterobacter sp.

Mercury (Hg) belongs to one of the noxious non-radioactive heavy metals, which is widely dispersed in nature [34]. Mercury is a persistent and bio-accumulative compound. Many polluted sites are co-contaminated with organic and metal pollutants. Data from both aerobic and anaerobic systems show that biodegradation of the organic compound may be reduced by metal toxicity [31,35,36,37]. Widespread use of pesticides and fertilizers may also have led to the presence of heavy metals in soil. The effect of mercury on the degradation of carbofuran was determined using 0.1 to 1 mg/L concentration. The outcomes revealed that concentrations of mercury from 0 to 1 mg/L had affected the degradation performance of carbofuran, as shown in Figure 3. Increasing the concentration of mercury to 0.5 mg/L caused a significant reduction in the degradation of carbofuran by the immobilized cells, which could be classified as detrimental as it influenced the degradation of the pesticides in soil by affecting the activity of microorganisms. An analysis of variance showed no significant difference (*p* ≥ 0.05) for immobilized cells between 0.1, 0.2, 0.3, and 0.4. It has been reported that metals appear to affect the biodegradation of organic compounds by affecting both the physiology and ecology of organic compound-degrading microorganisms [38]. Reports by Radjendirane et al. [37] showed that mercury could hinder hydroxylases activity like aryl 2,4-dichlorophenol hydroxylase and 3-hydroxybenzoate-6-hydroxylase. In another study by Ibrahim et al. [35], the authors found that concentrations of mercury from 0.1 to 1 mg/L affected the degradation of caffeine by *Leifsonia* sp. isolated from soil. The degradation of dibenzofuran by *Sphingomonas wittichii* RW1 was strongly inhibited by mercury at 1 mg/L [33].

### 2.5. Reusability of Gellan Gum Beads

The breakdown of the beads was observed when the beads began to disintegrate into semicircular parts, while some of the beads were scratched before overflowing. This might be a result of the mechanical force (shaking) that had been applied in the course of incubation using a shaker at 150 rpm. The bursting might have also triggered by the excessive growth of the entrapped cells, which could ultimately break the bead surface and cause leakage of cells. Figure 4 displays the results for the reusability of immobilized *Enterobacter* sp. strain BRC05 for degrading 100 mg/L carbofuran. Based on the result obtained, carbofuran was completely degraded at the first cycle, which was 24 h, and the carbofuran-degrading ability disappeared after reusing the immobilized cells for nine cycles. The result pointed out that immobilized bacterial cells could be reused for up to nine sequential complete degradation cycles without any decline in carbofuran-degrading ability [39].

Repeated application of remediation agent in the process of biodegradation is one of the benefits of the immobilized cells [24,40]. Hence, it was essential to find out how many times the immobilized cells produced in this study could be reused. The reusability experiment for immobilized bacterial cells was performed to study the viability of the entrapped cells in the degradation of carbofuran after one complete cycle. Manohar et al. [41] reported that beads containing entrapped *Pseudomonas* species were capable of degrading naphthalene 18 times in a cycle of 2–3 days before they lost their degrading activity. *Acinetobacter* sp. was used repeatedly up to five times. Ahmad et al. [24] revealed that the cells of immobilized *Acinetobacter* sp. were used for up to 47 successive cycles. The immobilized *Acinetobacter* sp. strain XA05 and *Sphingomonas* sp. were also recycled for up to 20 times reusability [42]. The reusability of beads (repeated batch) is one of the benefits of bacterial cell immobilization by which immobilized beads can be used for a prolonged period [43]. The reusability of beads means that the cells can survive, and metabolic activity can be preserved for prolonged periods. The reusability assessment was conducted on the immobilized *Enterobacter* sp. strain BRC05 cells in gellan gum to determine repeated usage of the beads for the degradation of carbofuran, as shown in Figure 5.

Massalha et al. [44] reported that an immobilized phenol-degrading microbe was recycled for three repeated cycles. There is inadequate literature on more than 50 cycles of reusability. An immobilized cell of *Klebsiella* sp. strain ATCC13883T could be recycled for more than 28 cycles in alginate-bentonite clay-PAC without losing any degradation capacity [21]. Immobilized *Burkholderia cepacia* PCL3 could be reused three times and still maintain its ability to degrade carbofuran. Based on the results obtained, it could be concluded that gellan gum immobilization had increased cell survival and metabolic activity in the bioremediation system.

## 3. Material and Methods

### 3.1. Chemicals and Medium

Analytical grade insecticide carbofuran (Furadan) 2,3-dihydro-2,2-dimethyl-7-benzofuranyl-methyl carbamate (99% purity) was purchased from Sigma-Aldrich, Missouri, USA. Minimal salt medium (MSM) and Luria Bertani (LB) contained the following compositions (in g/L): mineral agar 20 g, yeast extract 5 g, tryptone 10 g, NaCl 10 g, and distilled water 1000 mL. Then, 0.1 g/L of cycloheximide was added to all solid media to quash the growth of other microbes like fungus [45].

### 3.2. Analytical Technique

The standard solutions of carbofuran were prepared in acetonitrile to the final concentrations. The degradation of carbofuran was checked using HPLC (Agilent Technologies 1200 series, Waldbronn, Germany) using a column C18 ZORBAX^®^ 25 cm × 4.6 mm, fitted with a C18 silica reverse-phase guard column and equipped with a UV detector [46]. Carbofuran was detected at 254 nm wavelength at a run time of 5 min as well as the flow rate of 1 mL/min using HPLC grade acetonitrile (70%) and degassed water (30%) as the mobile phases. A sample amount of 20 μL was injected into the machine at an oven temperature of 26 °C. An external standard method was used for calibration. For quality control, the precision of the methods used in this study was established by HPLC injections of the samples in triplicate. The accuracy of the method was also ensured by running blank solvents and standards (every six injections) between the injections [6,47].

### 3.3. Sampling

Soil samples were collected from the agricultural soil with a history of pesticide application from vegetable plantation farms, in peninsular Malaysia. Ten composite soil samples (8–15 cm) from each sampling station were taken randomly using stainless steel scoop; the soil samples were collected in pre-sterilized bags and instantly sealed. The soil samples were stored at 4 °C to preserve the biological activity of the soil microorganisms [48].

### 3.4. Isolation of Carbofuran-Degrading Bacteria

Indigenous bacteria were isolated from the soil via enrichment methods. One gram of each soil sample was suspended in 10 mL of sterile mineral salts medium (MSM) containing an initial 5 mg/L of carbofuran as the sole carbon source. The 10-fold serial dilutions of the soil samples were inoculated into 100-mL MSM added with 50 mg/L carbofuran in the 250-mL flask, and the experiment was done under aseptic conditions in triplicate. A flask containing similar inoculum without the addition of carbofuran served as the control experiment. A 0.2 mL aliquot of the incubated and serially diluted enriched culture was poured in agar plates containing 50 mg/L carbofuran and then incubated at 37 °C for 48 h [49]. A sterile wire loop was used to pick the bacterial colonies in the plates and then restreaked into LB agar plates and incubated at 32 °C for two days. Sub-culturing was done every two weeks on carbofuran-containing agar plates until pure colonies were obtained [50]. The isolate was identified using biochemical tests and the 16S rRNA gene sequence.

### 3.5. Immobilization of Bacterial Strain

Bacterial cells were entrapped in gellan gum following an earlier described method by Ahmed et al. [24,51]. A culture was grown on a large-scale to obtain enough cells of the strain to be immobilized. The bacterial cells were provided with adequate oxygen through an oxygen pump to increase the aeration for the cells in the flask and speeding the growth of the cells. After 2 days of incubation, the culture was then centrifuged at 12,000× *g* for 10 min. The pellets obtained were used for cell immobilization experiments using gellan gum. About 0.75% (*w*/*v*) of gellan-gum was firstly added to a 100 mL distilled water and boiled to about 75 °C to completely liquefy the gellan gum. Then, 0.06% (*w*/*v*) CaCl_2_ was added to the solution and slowly chilled to nearly 50 °C at pH 7.0. The solution temperature was reduced to approximately 45 °C. Then, the subsequent bacterial pellet was added to the gum mixture while shaken continuously. An amended tip was used for the formation of beads by dropping the gum mixture into a sunflower canola oil containing 0.16% Span 80, which served as an emulsifier [22,52]. The equivalently sized beads were then separated from the oil into 500 mL of 0.2% (*w*/*v*) CaCl_2_. After two hours, the beads were recurrently rinsed with 0.1% (*v*/*v*) tween 80 solution to get rid of the oil phase from the microbeads. The beads were kept overnight in distilled water at 4 °C before being harvested by filtration and used for the degradation of carbofuran experiments [22].

### 3.6. Comparison of the Biodegradation of Carbofuran by Freely Suspended and Immobilized Cells

The evaluation of the degradation of carbofuran between immobilized and free cells of the bacterial isolate was made using the same initial biomass concentration for both immobilized and freely suspended cells. The optimum number of beads was used to estimate the number of microbial cells required for the degradation of carbofuran using free cells. Approximately 4% of the isolate was inoculated in a 100 mL MSM medium [53]. Concentrations of 50–250 mg/L carbofuran was used for both the immobilized and freely suspended cells. Different carbofuran concentrations were added into 100 mL of MSM medium, and the immobilized and free cells were added in a separate conical flask and incubated in a rotary shaker at 150 rpm at 37 °C temperature [13,21]. One milliliter of the samples was removed at some intervals of time and tested for the degradation of carbofuran until complete or maximum degradation of carbofuran was achieved. MSM medium without the addition of bacterial culture served as the controls. The experiments were conducted in triplicate [54].

### 3.7. Effect of Heavy Metals on the Degradation of Carbofuran by Enterobacter sp. BRC05

The effect of seven (7) different heavy metals was tested on the biodegradation of carbofuran by *Enterobacter* sp. Strain BRC05. The heavy metals used included copper (Cu), chromium (Cr), mercury (Hg), cadmium (Cd), arsenic (As), zinc (Zn), lead (Pb), and nickel (Ni) in 1 mg/L concentrations, which were preferred based on previous data on heavy metal concentrations in the study areas [3]. Thirty-six hours old bacterial culture with an optical density of 0.7–0.8 was used as inoculum for free cells and immobilized beads. The bacteria (free or immobilized) were inoculated into the MSM medium supplemented with 1 mg/L of each heavy metal. Medium without the addition of heavy metals was used as control [55]. The culture was incubated using a shaker incubator (150 rpm) at 37 °C. Bacterial growth was measured by taking the OD_600_ for free cells. The heavy metals that affected the degradation of carbofuran were observed, and the experiments were repeated for immobilized cells, and the incubation period was extended until the maximum degradation of carbofuran had been realized. All experiments were carried out in triplicates. 

### 3.8. Reusability of Gellen Gum Beads

In this study, 250 beads were added to 100 mL of medium supplemented with 100 mg/L of carbofuran. The cells were then incubated in a shaker at 150 rpm for 24 h, and then the residual carbofuran was measured during this period. After every 24 h incubation, the medium was thrown out, and the gellan gum beads were carefully washed and rinsed with distilled water before they were placed into a new fresh carbofuran medium. The steps were repeated at 24 h cycles until a reduction in the ability of the immobilized cells to degrade carbofuran was observed, and the beads started to disintegrate. All experiments were carried out in triplicates [17].

## 4. Conclusions

Local soil bacteria were isolated from agricultural farms in Malaysia and identified as *Enterobacter species* by using 16S rRNA sequencing. The cells of the bacterium were immobilized using gellan gum as an entrapment matrix. The outcome of the analysis showed that both free and immobilized cells of the bacterium were able to degrade carbofuran insecticide. Nevertheless, the performance of the immobilized cells was more efficient compared with the free cell system because the immobilized cells remain viable for longer periods and can withstand higher concentrations of carbofuran. Immobilization using gellan gum is a convenient technique where maximum cells remain viable and can withstand a high concentration of pollutants for a long time. Thus, the study showed the potential of immobilized *Enterobacter* sp. strain BRC05 in the remediation of soils contaminated with carbofuran. Furthermore, gellan gum is nontoxic to the bacterial cell, and it has low production cost and can be reused. The reusability saves the need for generating new biomass for each use. Moreover, the immobilized cells can be beneficial as a bioremediation agent in carbofuran contaminated sites. Furthermore, there is a need to further purify the carbofuran-degrading enzyme in future studies.

## Figures and Tables

**Figure 1 molecules-25-02771-f001:**
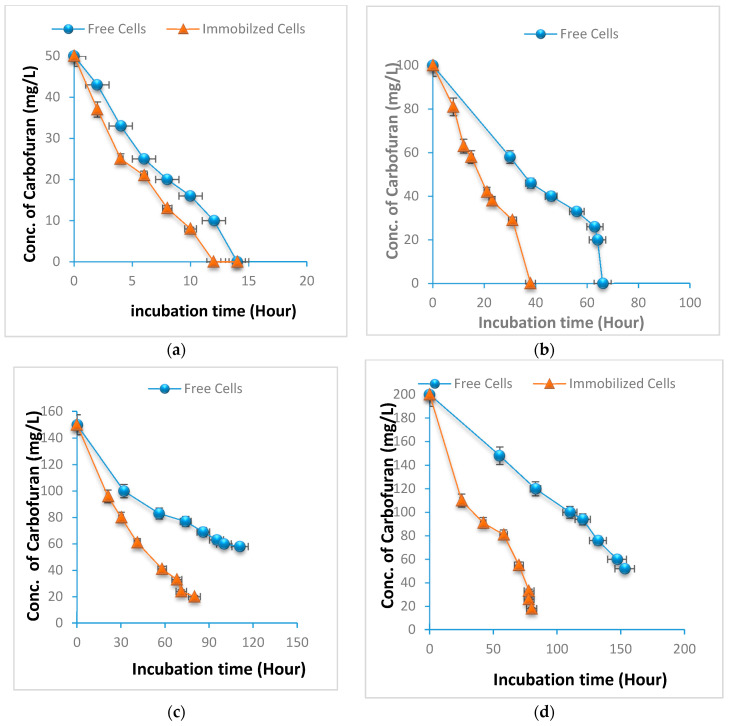
Effect of different concentrations of carbofuran on free and immobilized *Enterobacter* sp. cells on the degradation of carbofuran over a long period of incubation. The concentrations of carbofuran selected were (**a**) 50 mg/L, (**b**) 100 mg/L, (**c**) 150 mg/L, (**d**) 200 mg/L, and (**e**) 250 mg/L. Data represent mean ± SD, n = 3.

**Figure 2 molecules-25-02771-f002:**
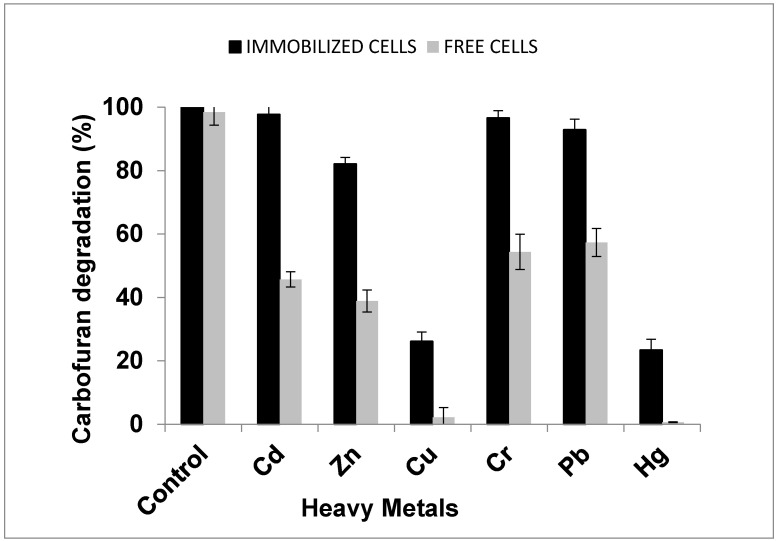
The effect of different heavy metals on the degradation of carbofuran by the immobilized and free cells of *Enterobacter* sp. Data represent mean ± STDEV, *n* = 3.

**Figure 3 molecules-25-02771-f003:**
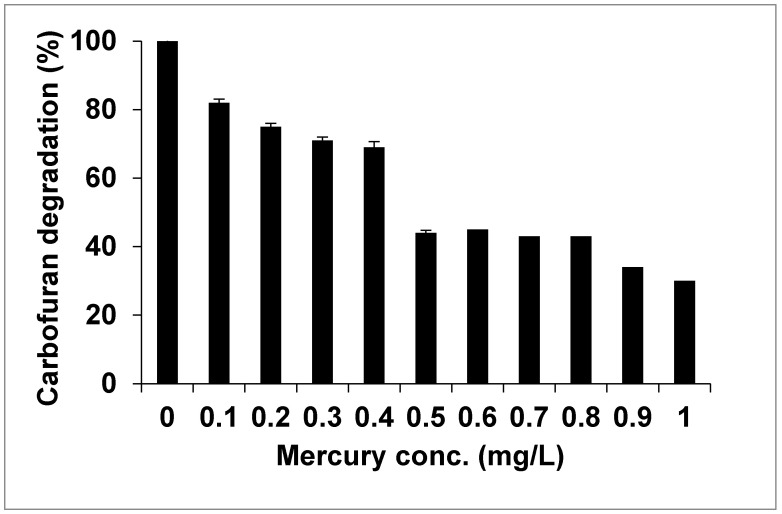
The effect of concentrations of mercury on the degradation of carbofuran by the immobilized cells of *Enterobacter* sp.

**Figure 4 molecules-25-02771-f004:**
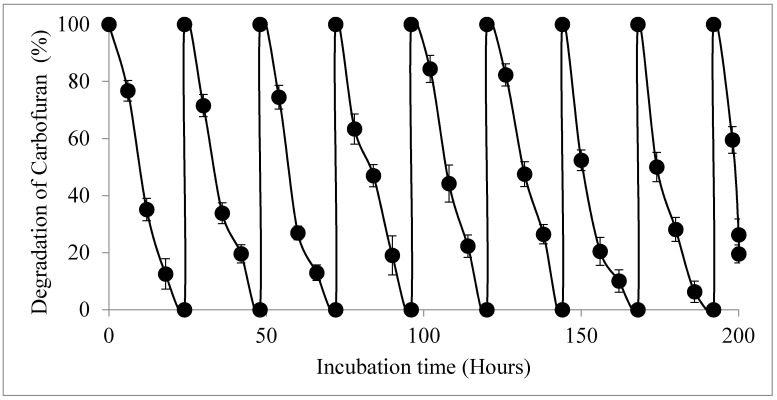
Repeated batch cycles of immobilized *Enterobacter* sp. Data is displayed for repeated usage only, and each cycle involved 24 h cycle. Data represent mean ± SD.

**Figure 5 molecules-25-02771-f005:**
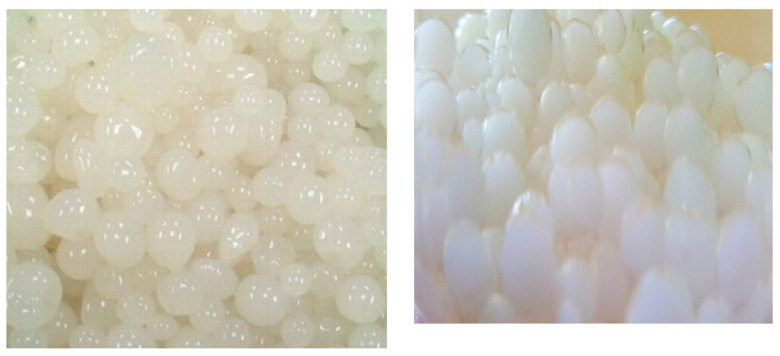
Gellan gum microbeads of *Enterobacter* sp. strain BRC05 before and after 24-h cycle degradation.

**Table 1 molecules-25-02771-t001:** The similarity of the isolate BRC05 16S rRNA gene sequence of the top ten Blast search result together with the 16S rRNA gene sequence of bacterium BRC05 and closely related species from the NCBI database.

Description	Max	Total	Query	E	Per.	Accession
	Score	Score	Cover	Value	Ident	
*Enterobacter cloacae* strain PSMK 16S ribosomal RNA gene, partial sequence	1537	2796	99%	0	99.64%	**MK641315.1**
*Enterobacter cloacae* strain JE3 16S ribosomal RNA gene, partial sequence	1537	2709	96%	0	99.64%	**KY942149.1**
*Enterobacter cloacae* R11 DNA, complete genome	1535	22072	99%	0	99.53%	**CP019839.1**
*Enterobacter* sp. pp9c chromosome, complete genome	1535	22253	99%	0	99.53%	**GQ360072.1**
*Enterobacter* cloacae strain HK196 chromosome, complete genome	1535	22007	99%	0	99.53%	**CP17087.1**
*Enterobacter cloacae* strain DF3 chromosome complete genome	1535	22072	99%	0	99.53%	**MG774409.1**
*Enterobacter* sp. PXG11 chromosome, complete genome	1535	22138	99%	0	99.53%	**JQ396391.1**
*Enterobacter cloacae* strain DMKU-RP206 16S ribosomal RNA gene, partial sequence	1535	2788	99%	0	99.53%	**MF125281.1**
*Enterobacter cloacae* strain MR1 chromosome, complete genome	1535	22166	99%	0	99.53%	**KC999878.1**
*Enterobacter* sp. Z-16 chromosome, complete genome	1535	21991	99%	0	99.53%	**DQ363438.1**

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
