# Peer review of "Enhanced Carbofuran Degradation Using Immobilized and Free Cells of Enterobacter sp. Isolated from Soil"

_molecules, 2020, doi:10.3390/molecules25122771_

Round 1

Reviewer 1 Report

I think that the paper improved after revision but still I think that it should be edited by a native English speaker

Author Response

We thank the reviewer for his comments. We have reviewed carefully the entire manuscript and have removed redundancies, and the English was edited thoroughly in the revised manuscript.  

Reviewer 2 Report

This paper, entitled Enhanced carbofuran degradation using immobilized and free cells of Enterobacter sp. isolated from soil, is a scholarly work and can increase knowledge on this domain. The authors provide an interesting study, the content is relevant to Molécules. The paper is quite well written and well related to existing literature. The abstract and keywords are meaningful.

I have some specific and general comments;

  • why some sentences are highlighted in yellow? is it a revised version submitted after a first review?
  • from my point of view, the introduction section should be improved and enlarged, the authors should give more precision about the context and the objectives of this work. This section is too short and unsufficiently related to existing literature and previous studies dealing with same methodology and approach. What is the statement of this study according to other studies? What are the current state of the art on such approach? This point must be improved in depth.
  • Please discuss about methodologies of cells immobilization? What the main techniques? the advantages and limits of immobilization? Is it easily applicable at pilot scale or industrial scale? Please discuss this.
  • Why the choice of carbofuran? What is the main interest to consider this molecule? What are the associated risk? 
  • About soil samples, how many samples were analyzed? what is the repeatability of each experiment? duplicate? triplicate? What is the traceability of pesticide application on this soil? What are the amounts in this soil?
  • Why the choice of these heavy metals used in this study? Is it based on literature benchmark? on previous studies or experiments carried out by authors? Why this heavy metals? Are the effects already described in other studies?
  • Table 1 is dealing with results of molecular biology but there's no information in Materials and methods section...
  • The Results section is mainly descriptive… Discussion and interpretation of results are lacking in this section. Please discuss in depth the results and do not give only description of results. This section must be improved in depth also.
  • Please check legend of Figure (incubation instead of incubabation).
  • Why length of X axis is not the same for each figure?  it's not easy to compare and to visualize difference between each condition.
  • Please give costs analysis of such approach. Please give costs vs benefits discussion.
  • What are the next steps? What are the next experiments? Please discuss about perspective of this work.

As it, this paper is not acceptable for publication and requires at least a major revision to be considered for publication in Molecules. I recommend the following decision: RECONSIDER AFTER MAJOR REVISION.

Author Response

RESPONSE TO REVIEWER TWO COMMENTS

We thank the reviewers and the editors for their thoughtful comments and observations. We have revised our manuscript carefully in response to their suggestions and hope that this improved manuscript would be reviewed for possible publication in this reputable journal. We hope the revised version is now suitable for publication and look forward to hearing from you in due course. We have highlighted in yellow all the changes made in the main manuscript.

Comment

why some sentences are highlighted in yellow? is it a revised version submitted after a first review?

Response

We appreciate the time and effort that you and the reviewers have dedicated to providing your valuable feedback on this manuscript. Yes, the manuscript was a revised version resubmitted again.

Comments

from my point of view, the introduction section should be improved and enlarged, the authors should give more precision about the context and the objectives of this work. This section is too short and unsufficiently related to existing literature and previous studies dealing with same methodology and approach. What is the statement of this study according to other studies? What are the current state of the art on such approach? This point must be improved in depth.

Please discuss about methodologies of cells immobilization? What the main techniques? the advantages and limits of immobilization? Is it easily applicable at pilot scale or industrial scale? Please discuss this.

Response

The introduction section was improved and enlarged using previous studies dealing with same methodology and approach. The current state of arts in the area was discussed also methodologies and techniques, limitation and application of immobilization methods were discussed in the introduction section as suggested.

Comments

Why the choice of carbofuran? What is the main interest to consider this molecule? What are the associated risk? 

Response

Carbofuran was chosen for the study because consumers paid less attention about its toxicity and high potential for ground water contamination and has also been detected in surface waters. Carbofuran usage has increased in recent years’ despites its total banned because it is one of the few insecticides effective on many crop destroying insects. This situation calls for urgent attention with acceptable solution for the removal of the compound from the environment. Meanwhile, molecule acts as a platform for the authors to contribute their findings and help raise awareness among community on toxicity assessment and prevention.

Comments

About soil samples, how many samples were analyzed? what is the repeatability of each experiment? duplicate? triplicate? What is the traceability of pesticide application on this soil? What are the amounts in this soil?

Response

We thank the reviewer for the comments description about sampling, Soil samples were collected from different agricultural farms in peninsular Malaysia. The specific study areas are agricultural farms where vegetables like cabbage, rice, tomatoes, and Sweet Potatoes are cultivated and was choose because of prolonged history of different pesticides applications in those areas. The aim is to isolate bacteria capable of degrading carbofuran as for traces of pesticides in the soil samples the sampling areas were chosen based on previous reports on pesticides contamination in the sampling areas.  All experiments were performed under aseptic conditions in triplicate.

Comments

Why the choice of these heavy metals used in this study? Is it based on literature benchmark? on previous studies or experiments carried out by authors? Why this heavy metals? Are the effects already described in other studies?

Response

Mostly the pesticide polluted sites are co-contaminated with heavy metals. Thus, pesticide biodegradation might be affected by the existence of heavy metals commonly found in the environment. Heavy metals are naturally occurring compounds, however, they are frequently introduced via anthropogenic activities into the environments many previous literatures reported the effect of heavy metals at the concentrations used 1 mg/L as cited in the manuscript.

Comments

Table 1 is dealing with results of molecular biology but there's no information in Materials and methods section...

Response

The table is showing the relationships between isolated species and other closely related strains which closely matches the isolated strain Enterobacter sp. strain BRC05 was retrieved from NCBI Gene Bank and was describe briefly under section 2.4 Isolation of carbofuran degrading bacteria.

Comments

The Results section is mainly descriptive… Discussion and interpretation of results are lacking in this section. Please discuss in depth the results and do not give only description of results. This section must be improved in depth also. Please check legend of Figure (incubation instead of incubabation).

Why length of X axis is not the same for each figure?  it's not easy to compare and to visualize difference between each condition.

Response

We thank the reviewer for the comments description about results were elaborated. Legend of Figure (incubabation) was corrected and the lengths of axis were adjusted as suggested

Comments

What are the next steps? What are the next experiments? Please discuss about perspective of this work.

Response

The application of immobilization technology to environmental area is in its preliminary stages, but the results seen so far are promising. The immobilized cells will be useful to treat the waste to convert the toxicant into nutrient, biomass and CO2 via biodegradation through their intermediates. Better biodegradation rate was observed in immobilized cells due to absence of internal and external mass transfer resistance. Our next experiments are to focus on microcosm study using the strain to see its capability in field condition

Round 2

Reviewer 2 Report

The authors provide now a revised version of their manuscript taking into account the comments made by reviewers. There's again some points to improve especially about the Figure 1. Please check all the legends (immobilized for example, there's a mistake in one) and please plot the figures with same scale for X and Y axis, or maybe combine all curves in one figure. I see no difference between Fig1C and Fig1D, it seems to be the same.

Please check the manuscript and modify this point , that is important to solve I think. I recommend the following decision: RECONSIDER AFTER MAJOR REVISION.

Author Response

Response to reviewers comments 

We thank the reviewer for his comment. We have checked the figures 1C and 1D again we have placed the figures in their appropriate area in the manuscript. We are very sorry for this mistake.

This manuscript is a resubmission of an earlier submission. The following is a list of the peer review reports and author responses from that submission.

Round 1

Reviewer 1 Report

In general the manuscript is interesting, is clearly presented and the design of the experiments are appropriate; but in my opinion authors might described with more detail the assays of biodegradation not only in method but also in results.

However; I have found a big mistake. I always thought that the genus  Enterobacter is a Gram negative rod belonged to Family Enterobacteriaceae. Motile by peritrichum flagellum. oxidase negative and it is a non spore forming bacterium. 
This mistake must be corrected

I attach the revised version of the manuscript

Author Response

We thank the reviewers for their thoughtful comments. We have revised our manuscript in response to their suggestions and hope that this improved manuscript.

Reviewer 2 Report

The topic is potentially promising, however several issues should be resolved by the Authors to ensure that the manuscript meets the scientific standards of Molecules. The issues were listed below:

  1. There is no aim of the study. The final paragraph of the INTRODUCTION section should always include a description of the scientific goal or the research hypothesis that the Authors wanted to achieve/test throughout their experiment. This is necessary, as otherwise the Reader cannot properly evaluate the outcome of the study.
  2. In the MATERIALS AND METHODS section, the description of media (section 2.1) seems to lack data and is rather confusing. Furthermore, the details regarding the procedure used for identification of the isolate should also be placed in this section. Please revise it accordingly to ensure that the experiment may be repeated.
  3. RESULTS AND DISCUSSION section – some structural changes should be employed. Why is the description of isolation and identification results not listed as a separate sub-section? It should be placed as 3.1. I would also recommend to substitute the following sections – it is logical that the biodegradation results should be presented first (as sub-section 3.2), whereas the reusability should be described afterwards (sub-section 3.3).
  4. RESULTS AND DISCUSSION section – please provide the results obtained for negative controls in respective graph of fig. 4. According to the description in the MATERIALS AND METHODS section, the sample for analysis of carbofuran content was collected directly from the culture – it is therefore possible that the depletion of carbofuran concentration may result from sorption to the carrier/immobilized cells. This aspect is often omitted and is a common source of conceptual errors. In order to exclude this issue, the results for negative controls should be presented.
  5. RESULTS AND DISCUSSION section – the discussion part (lines 229 – 251) is very general and descriptive. It should include direct numerical values and comparisons in order to establish whether the solution presented by the Authors is valuable and interesting. Furthermore, the discussion should consider how the results presented by the Authors extend the current state of the art (where is the novelty?), state the limitations of the study (e.g. I imagine that different results would be obtained in a soil system) and highlight future considerations (what should be investigated in future studies?).
  6. CONCLUSION section – again, very general and speculative. The statement that “the performance of immobilized cells of the bacterium was more efficient compared to the free cell system” is a well-known fact – this is rather typical and there is no novelty in this discovery. The Authors claim that “the immobilized cells can be beneficial as a bioremediation agent in carbofuran and other pesticides contaminated sites” – the part regarding other pesticides is not supported by data. Please re-write this section, with more emphasis on the novelty and practical applicability.
  7. The manuscript requires severe editing as there are numerous syntax, grammar and style issues. I would strongly recommend to consult a native speaker. Some examples were listed below:

Gramar errors:

  • its, toxicity and persistence nature
  • Also, it is one of the major contributors to groundwater contamination due to their mobility
  • has made them among the
  • standards solutions of carbofuran was prepared
  • Carbofuran concentrations ranging from 50 –250 mg/L was used
  • Figures 4 (a-e) displays
  • Indigenous soil bacteria was isolated

Syntax errors:

  • An isolate BRC05 isolated from agricultural soil was characterized and presumptively identified as Enterobacter sp. was immobilized
  • Though carbofuran was chemically not stable due to its easy
  • disruptor and has one of the highest acute toxicities to humans of any pesticide broadly applied field crops.
  • a favorable strategy in most process of bioremediation technology
  • Carbofuran degradation of was done using HPLC

Typos:

  • The immobilize cells may be recycled
  • application couple with solubility in water
  • Nazi et al., (2001), reported that beads containing entrapped Pseudomonas specieswereablycapable

Editing:

  • The names of bacterial species should always be italicized (lines 189 - 251).

Author Response

RESPONSE TO REVIEWER 2 COMMENTS

We thank the reviewer for the thoughtful comments. We have revised our manuscript in response to the suggestions and comments of the reviewer hope that this improved manuscript is acceptable for publication.

  1. Comment

There is no aim of the study. The final paragraph of the INTRODUCTION section should always include a description of the scientific goal or the research hypothesis that the Authors wanted to achieve/test throughout their experiment. This is necessary, as otherwise the Reader cannot properly evaluate the outcome of the study.

  • Response

Thanks for this suggestion the aim of the study was included in the final paragraph of the introduction section as suggested.

  1. Comment

In the MATERIALS AND METHODS section, the description of media (section 2.1) seems to lack data and is rather confusing. Furthermore, the details regarding the procedure used for identification of the isolate should also be placed in this section. Please revise it accordingly to ensure that the experiment may be repeated.

  • Response

Thanks for the comments, in almost every biotechnological or experimental research a chemical and medium is used in one form or another some are purified while others received without any further purification. Therefore, it is important to explain the purity, grade and the purchasing company for easy evaluation of outcomes from the instruments used as clarity on grades of chemical reagents helps achieve high precision of analysis as well as reference for other researchers. I would also like to state that the study is part of a large project the author has conducted a molecular 16S rRNA to identify the organism. However, the author have provided the top ten Blast search result closely related species to BRC05 strain an Enterobacter sp.

  1. Comment

RESULTS AND DISCUSSION section – some structural changes should be employed. Why is the description of isolation and identification results not listed as a separate sub-section? It should be placed as 3.1. I would also recommend to substitute the following sections – it is logical that the biodegradation results should be presented first (as sub-section 3.2), whereas the reusability should be described afterwards (sub-section 3.3).

  • Response

We thank the reviewer for this positive comments and suggestions the heading isolation and identification results was listed as separate sub-section and biodegradation results was presented first (as sub-section 3.2), then reusability afterwards as (sub-section 3.3).

  1. Comment

RESULTS AND DISCUSSION section – please provide the results obtained for negative controls in respective graph of fig. 4. According to the description in the MATERIALS AND METHODS section, the sample for analysis of carbofuran content was collected directly from the culture – it is therefore possible that the depletion of carbofuran concentration may result from sorption to the carrier/immobilized cells. This aspect is often omitted and is a common source of conceptual errors. In order to exclude this issue, the results for negative controls should be presented.

  • Response

We thank the reviewer for their careful reading of the manuscript and their constructive remarks. As a general rule, you need a negative control to validate a positive result, similarly, in the present study, a negative control was used which is empty beads without bacteria used at the same carbofuran concentrations. And no degradation observed in the negative control beads at the end of the experiment and that validate the experiment in in respect of the graph of fig. 4.

  1. Comment

RESULTS AND DISCUSSION section – the discussion part (lines 229 – 251) is very general and descriptive. It should include direct numerical values and comparisons in order to establish whether the solution presented by the Authors is valuable and interesting. Furthermore, the discussion should consider how the results presented by the Authors extend the current state of the art (where is the novelty?), state the limitations of the study (e.g. I imagine that different results would be obtained in a soil system) and highlight future considerations (what should be investigated in future studies?).

  • Response

We thank the reviewer for this positive comments, we have elaborated the results and discussion part (lines 229 – 251) and revised. We also compared with recent studies conducted in the area. future considerations was highlight in the conclusion section.

  1. Comment

CONCLUSION section – again, very general and speculative. The statement that “the performance of immobilized cells of the bacterium was more efficient compared to the free cell system” is a well-known fact – this is rather typical and there is no novelty in this discovery. The Authors claim that “the immobilized cells can be beneficial as a bioremediation agent in carbofuran and other pesticides contaminated sites” – the part regarding other pesticides is not supported by data. Please re-write this section, with more emphasis on the novelty and practical applicability.

  • Response

We appreciate the reviewer’s suggestion for a more specific conclusion to support the presented data meanwhile, the part regarding other pesticides was rewrite and revised also practical applicability was highlighted as suggested.

  1. Comment

The manuscript requires severe editing as there are numerous syntax, grammar and style issues. I would strongly recommend to consult a native speaker. Some examples were listed below:

Gramar errors:

  • its, toxicity and persistence nature
  • Also, it is one of the major contributors to groundwater contamination due to their mobility
  • has made them among the
  • standards solutions of carbofuran was prepared
  • Carbofuran concentrations ranging from 50 –250 mg/L was used
  • Figures 4 (a-e) displays
  • Indigenous soil bacteria was isolated

Syntax errors:

  • An isolate BRC05 isolated from agricultural soil was characterized and presumptively identified as Enterobacter sp. was immobilized
  • Though carbofuran was chemically not stable due to its easy
  • disruptor and has one of the highest acute toxicities to humans of any pesticide broadly applied field crops.
  • a favorable strategy in most process of bioremediation technology
  • Carbofuran degradation of was done using HPLC

Typos:

  • The immobilize cells may be recycled
  • application couple with solubility in water
  • Nazi et al., (2001), reported that beads containing entrapped Pseudomonas specieswereablycapable

Editing:

  • The names of bacterial species should always be italicized (lines 189 - 251).

  • Response

We thank the reviewers for their time and constructive comments all grammatical, syntax and typing errors were corrected and revised . 

Reviewer 3 Report

The subject of the paper is interesting but I think that the paper should be rewritten and edited by a native English speaker before its scientific merit could be evaluated, because it is difficult to follow, for example section 2.

The experimental details should be given, for example concentrations, pH (this a very important parameter and the initial and equilibrium pH should be given), etc.

Is the carbofuran stable in the experimental conditions?

Author Response

Response to Reviewer 3 Comments

We appreciate the time and effort the reviewers dedicated to providing feedback on our manuscript and we are grateful for the insightful comments and valuable suggestions to our paper. We have incorporated most of the suggestions made by the reviewers within the manuscript.

Comments

The subject of the paper is interesting but I think that the paper should be rewritten and edited by a native English speaker before its scientific merit could be evaluated, because it is difficult to follow, for example section 2.

Response

We thank the reviewer for this thoughtful comments the section 2 of the manuscript was thoroughly checked especially for English editing

Comment

The experimental details should be given, for example concentrations, pH (this a very important parameter and the initial and equilibrium pH should be given), etc. Is the carbofuran stable in the experimental conditions?

Response

The pH of the medium was adjusted to neutral and carbofuran is Stable under neutral or acid conditions. However, it is unstable in alkaline media.

Round 2

Reviewer 3 Report

I still think that the subject of the paper is interesting but I think that the paper should be rewritten and edited by a native English speaker before its scientific merit could be evaluated.

Some comments

Table 1 is not mentioned in the text, it should be explained.

There is not figure 1 in the manuscript.

I suppose that figure 4 is figure 1.

The experimental details should be given.

In general the paper should be revise.

Author Response

Dear editor,

RESPONSE TO REVIEWERS COMMENTS

We thank the editors for their thoughtful comments. We have revised our manuscript in response to their suggestions and hope that this improved manuscript would be reviewed for possible publication in this reputable journal.

Comment 1:
Table 1 is not mentioned in the text, it should be explained.

Response 1

We thank the reviewer for this important observation. The author has explained and mention table 1 in the text

Comment 2

There is not figure 1 in the manuscript.

Response 2

Thanks for the list of figures were revised in the manuscript.

Comment 3.

I suppose that figure 4 is figure 1.

Response 3

Yes, all the figures in the manuscript were rearrange and revised.

Comment 4

The experimental details should be given.

Response 4

We appreciate the reviewer’s suggestion the experimental details of how the beads were formed and compound detection methods were already presented in the manuscript and revised.

Comment 5

In general, the paper should be revise.

Response 5

Thanks for the comments, we have carefully revised the manuscript as the recommended.